# Long-Term Use and Application of Systematic Tailored Assessment for Responding to Suicidality (STARS) Protocol Following Original Training

**DOI:** 10.3390/ijerph191811324

**Published:** 2022-09-08

**Authors:** Jacinta Hawgood, Kairi Kõlves, Susan H. Spence, Ella Arensman, Karolina Krysinska, Diego De Leo, Tamara Ownsworth

**Affiliations:** 1Australian Institute for Suicide Research and Prevention, World Health Organization Collaborating Centre for Research and Training in Suicide Prevention, Griffith University, Brisbane, QLD 4122, Australia; 2School of Public Health, College of Medicine and Health, University College Cork, Western Gateway Building, T12 XF62 Cork, Ireland; 3National Suicide Research Foundation, University College Cork, Western Gateway Building, T12 XF62 Cork, Ireland; 4Centre for Mental Health, Melbourne School of Population and Global Health, The University of Melbourne, Melbourne, VIC 3010, Australia; 5School of Applied Psychology, The Hopkins Centre, Menzies Health Institute Queensland, Griffith University, Brisbane, QLD 4222, Australia

**Keywords:** suicide risk assessment, the STARS-p, adherence, fidelity, structured professional judgement

## Abstract

Background: Understanding the use of Systematic Tailored Assessment for Responding to Suicidality protocol (STARS-p) in practice by trained mental health practitioners over the longer- term is critical to informing further developments. The study aim was to examine practitioners’ experiences of STARS-p and factors associated with its use in practice over a 12–24-month period after training. Method: Practitioners who undertook the STARS-p training completed an online survey 12–24 months post training. The survey focused on the frequency of use of STARS-p (in full and each section) as well as perceptions about STARS-p applied in practice. Analyses included correlations, logistic regression and content analysis. Results: 67 participants (81% female, Mage = 43.2, SD = 10.3) were included in the analyses. A total of 80.6% of participants had used the entire STARS-p at some time-point in their practice and less than half (44.7%) frequently used the entire STARS-p (all components in one administration). Parts A, B and C were used frequently in suicide risk assessment (SRA) by 84%, 71% and 82% of participants, respectively. Use of the entire protocol and different sections was most related to male gender, perceived ease of administration and confidence in the use of the protocol. Qualitative results revealed three main themes. Conclusions: STARS-p as a whole or its parts, is frequently used. Advantages of, and barriers to, using STARS-p in practice can inform further developments of STARS-p and STARS training.

## 1. Introduction

Since the early to mid-1990s, national suicide prevention strategies have been implemented throughout the world in response to increasing rates of suicide and non-fatal suicidal behavior [1]. Given the potential for negative mental health impacts due to the COVID-19 pandemic, there has been increased focus on suicidality [2,3]. Some nations have responded with increased funding for resources for those at risk of suicide [4]. For example, the Australian government introduced a job keeper employment support program and additional mental health funding, expanding better access to mental health services [5]. Additional innovative digital health care models have also been proposed, including preclinic triage assessment and care systems designed to improve the appropriateness and effectiveness of referrals [6]. An emphasis on suicide risk assessment (SRA) and monitoring for those at heightened risk of suicide is strongly advocated for reducing suicide [7]. In Australia, the Fifth National Mental Health and Suicide Prevention Plan [8] promotes integrated planning and service delivery for suicide prevention and MH services from 2017 to 2022. Further, the Final Advice report by the Federal Government [9] identifies ‘workforce and community capability’ as an enabler for suicide prevention (see Recommendation 4, Department of Health, 2020).

Evidence-based training, which enables mental health practitioners (MHPs) to respond competently and compassionately to those in suicidal distress, is a key focus of this recommendation. Equipping MHPs with evidence-based SRA training is one means of creating a competent suicide prevention workforce [10]. Thus, training is important for enhancing competencies of gatekeepers [11], in addition to developing systematic capabilities related to SRA in the MH workforce [10,12,13].

### 1.1. Suicide Risk Assessment Processes—Structured Professional Judgement and Application

It has been widely acknowledged that utilizing SRA to predict who will engage in suicidal behavior is problematic [14,15,16,17,18]. Research has shown that SRA tools suffer from very low positive predictive values (PPVs) in the range of 1.3 to 16.7% with 87% false positives [19,20], with suicide prediction estimates not significantly better than chance [21]. Consequently, the use of SRA for prediction has shifted to a focus on identifying psycho-social needs of a person in suicidal distress, to inform appropriate needs-based management practices [22].

Interview-based methods which use structured professional judgement are becoming best practice [23,24] for SRA focused on identification of psycho-social needs. Structured professional judgement combines empirical data on psycho-social risk and protective factors, clients’ subjective experiences of these, and a systematic suicide state enquiry [10]. Further, existing SRA guidelines support a psycho-social needs-based understanding of suicidality [14,25], with associated methods found to reduce the risk of repeat suicide attempts, particularly for those without a history of psychiatric treatment [26,27], as well as enhancing patient compliance in aftercare [25].

Yet, little is currently known about how MHPs utilize SRA protocols or apply structured professional judgement in practice. One study was identified on feasibility and implementation fidelity of the Risk of Suicide Protocol (RoSP) [28]. The RoSP is a structured professional judgement approach which is designed to evaluate suicidality and inform safety planning in line with the National Institute for Health and Care (NICE) Guidelines for SRA developed in the United Kingdom [29]. A recent evaluation of the efficacy of RoSP showed that it is a reliable and valid instrument for evaluation of suicidality in clinical and community mental health services [24]. However, clinician perceptions of the use and uptake of the RoSP in practice have yet to be examined. In fact, in the SRA literature generally, there is a lack of understanding of MHPs’ application of the protocols they have been trained to use and their views of these in practice.

We could identify only two studies investigating therapists’ applications of an SRA management and treatment framework following training [30,31].

### 1.2. Systematic Tailored Assessment for Responding to Suicidality

In Australia, an increasingly well-known structured professional judgement approach to SRA is the Systematic Tailored Assessment for Responding to Suicidality protocol (STARS-p) [32]. As outlined in Figure 1, the STARS-p includes empirically informed questions regarding indicators of suicidality (Part A), psycho-social needs/risk factors (Part B) and protective factors (Part C), to facilitate systematic exploration of a client’s experience of suicidality and their psycho-social context. STARS-p also includes client-rated ‘levels of concern’ associated with elements of the suicidal enquiry (Part A) and psycho-social needs/risk factors (Part B). The client’s key identified concerns (from Parts A and B), client reported protective factors (Part C) and clinician judgment are integrated to determine priority foci for safety planning (collaboratively developed), and immediate management responses (outlined in the clinical notes section).

### 1.3. Mental Health Professionals’ Perceptions of STARS-p and the Need for Training in Its Administration

Using the original version of STARS-p (2015 Edition: [33]), Hawgood and colleagues [34] investigated MHPs’ perceptions of ease and practicality of administration, client-centeredness, and confidence in the data for informing care responses. Perceptions of client-centeredness and confidence in the data were largely positive; however, suggestions for improving ease of administration were made, which included a recommendation that STARS-p training be made essential for MHPs prior to use of STARS. This feedback directly informed the updated STARS-p (2018 Edition) and the development of mandatory STARS-p training for those implementing STARS in practice. The capabilities required in the conduct of comprehensive SRA entails competence beyond that required for administration of SRA actuarial scales and gatekeeper training [13,35,36]. STARS-p training was developed to align with minimum standardized competencies in SRA, proposed by Cramer and colleagues [12], and in collaboration with those who have a lived experience of suicide (Lived experience of suicide has been defined as “having experienced suicidal thoughts, survived a suicide attempt, cared for someone through suicidal crisis, or been bereaved by suicide” (Roses in the Ocean, 2016 [37]).)

A recent evaluation of the short-term impacts of STARS-p training showed that it has a positive impact on capabilities of MHPs [10]. Specifically, a pre–post evaluation of the two-day training (N = 222) identified significant improvements in clinicians’ attitudes, perceived capabilities and declarative knowledge regarding suicide prevention. MHPs with less prior formal training, as well as those with fewer years’ experience, showed the greatest improvement in terms of perceived capability, and those with less informal training (e.g., supervision) pre-training were most likely to show improved attitudes regarding suicide prevention. However, to date, there has been no investigation of STARS-trained MHPs’ perceptions of use of the STARS-p in the longer-term after training, including frequency of administering the entire protocol, versus different sections, and perceptions of ease of administration, client-centeredness, and barriers to application in practice.

STARS training emphasizes the importance of administration of the entire STARS-p during assessment for optimizing client outcomes (i.e., administration of Parts A, B, C, safety planning and clinical notes in one session). Earlier findings showed most MHPs (57.9%) report variation and flexibility in the administration order of sections of STARS-p [34]. Further, factors such as time constraints and presentation of acuity of client suicidal distress may influence feasibility of administering the entire protocol (which takes approximately 1–1.5 h, including the safety planning and clinical notes documentation) [10].

## 2. Study Rationale and Aims

It is currently unknown how STARS-trained MHPs utilize STARS-p in the real-world with their clients over the longer-term (i.e., 12–24 months after training). Understanding MHPs’ perceptions of client-centeredness, use and effectiveness of the STARS-p (2018 Edition) and factors influencing uptake in practice are essential to inform the implementation of SRA for prevention of suicide.

Accordingly, the current study’s aims were to:Examine STARS-trained MHPs’ use, perceptions and reasons for use of STARS-p over the 12–24-month period post-training. It was hypothesized that, when using STARS-p, the majority of participants would not use the entire STARS-p with their clients in one administration. Further, it was expected that the main reason for administering STARS-p in practice would be client expressions of suicidality.Investigate associations between practitioners’ demographic and clinical characteristics and their use and perceptions of STARS-p in practice over the 12–24-month post-training period. It was hypothesized that greater formal training (workshops) and informal training (supervision) would be associated with more frequent administration of the entire STARS-p. Further, it was hypothesized that practitioners reporting that they typically administered the entire STARS-p would have more positive perceptions of ease of administration, client experience (of being validated), client-centeredness, as well as confidence in the usefulness of data for informing interventions to mitigate suicidality.

As an exploratory component, for which there were no hypotheses, a further aim was to identify STARS-trained MHPs’ perceptions of the STARS-p and suggestions as to how STARS-p can be improved. Specifically, their views about any challenges and improvements for enhancing use of the STARS-p were sought.

## 3. Materials and Methods

### 3.1. Participants and Recruitment

Participants were Australian MHPs (with and without formal qualifications) working with persons who experience suicidality (including psychologists, social workers, nurses, occupational therapists, counselors, welfare workers, youth workers, peer and support workers), and who undertook the STARS two-day training workshop during 2019–2020. They were followed up between 12 and 24 months post STARS training and invited to participate in this study. Out of the total STARS-trained invited participants (N = 165) on the STARS training database between 2019–2020, we received 43 returned emails indicating the person had left the workplace (no longer worked at the address). Therefore, of the 122 participants able to be contacted to participate in this study, 70 responded (57% response rate). This response rate was considered acceptable, and not unexpected, given the longer-term follow-up [38] and the high workforce attrition rate observed in the community mental health and suicide prevention workforce, especially during the COVID-19 pandemic [39].

Participants were classified into two groups, namely, MHPs with a formal MH qualification (MHP-wQ) and MHPs without a formal MH qualification (MHP-woQs; e.g., general counseling or welfare/youth workers). At follow-up, participants were sent a link to an online survey containing closed- and open-ended questions and offered a $25 AUD gift voucher as reimbursement for their time in completing the survey, as well as an option to enter the prize draw for one of 15 chances to win a $100 gift voucher.

All procedures were approved by the Griffith University Human Research Ethics Committee (Ref number: 2015/813/HREC). Surveys were set up in Research Electronic Data Capture (REDCap), a secure online instrument, which was opened for 4 months from July to October 2021. Three email reminders sent within a month followed the original email invitation. All participants indicated consent by proceeding with the online survey.

### 3.2. Questionnaire

Demographic and work background information regarding age, gender, practitioner role, education, years in suicide prevention, SRA training and supervision, and, more specifically, the amount of formal training (e.g., workshops) and informal training (supervision/mentoring), and experience of client suicide and/or suicide attempt was collected.

#### Use of STARS-Protocol and Clinician Perceptions

The following survey sections appeared in the online questionnaire in the following order:

*Reasons for administering STARS-p*: A 13-item list of reasons for administering the STARS-p at intake or during a session with a client was included. Example items included: ‘direct verbalization of suicidal ideation’, ‘observation of mental illness symptoms’, ‘reported psycho-social stressors’, and ‘mention of suicidality in a referral/presenting problem’. There was also an option for participants to note whether they ‘always conduct a risk assessment (given that), it is mandatory in my organization’.

*Use of STARS-p administration*: Participants were asked how frequently they administered the entire STARS-p (i.e., all sections of the protocol as per the STARS training guidelines) when they undertook a suicide risk assessment with a client. The survey commenced with, “We understand that there may be some occasions in which only select parts of the STARS-p are used, please indicate your frequency of use for each part.” Frequency of administration of the entire STARS-p and each section was measured via a 5-point Likert scale (1 = 100% of the time, 2 = 75% of the time, 3 = 50% of the time, 4 = 25% of the time, and 5 = Never).

*Perceptions of the STARS-p*: Participants were asked to rate their perceptions of five attributes of the STARS-p on a 5-point Likert scale, including ‘ease of administration’ (1 = extremely difficult to 5 = extremely easy); ‘perceived client experience’ (1 = very invalidated/misunderstood to 5 = very validated/understood); ‘confidence in general usefulness of data obtained’ (1 = not at all confident to 5 = very confident); ‘confidence in use of data for informing needs-based priority areas for interventions’ (1 = not at all confident to 5 = very confident); and perceptions of STARS as a client-centered approach for assessing current suicidality (1 = not at all client-centered to 5 = very client-centered).

Two final questions were designed to gain participants’ perceptions of use in practice and suggestions for future STARS-p or STARS training modifications: (1) Any further comments about how you use STARS-p? and (2) Any further comments or suggestions regarding the use of STARS-p or STARS training?

### 3.3. Analyses

Descriptive analyses were conducted to derive means with standard deviations and frequencies. For the correlational analyses of demographic and clinical factors related to clinicians’ use and perceptions of the STARS-p, Spearman’s correlation was used. Further analysis was conducted using multiple logistic regression to identify the independent factors related to the use of STARS. Considering the ordinal nature of most of the variables and modest sample size, the variables were dichotomized. The outcome measures (dependent variables) were the use of STARS-p; 75% use was utilized as a cut-off point (i.e., 100% and 75% considered as ‘frequent use’ and never to 50% as ‘infrequent use’). Forward stepwise likelihood ratio models were used to include all significant variables from the correlation analyses. Odds ratios (ORs) with a 95% confidence interval (CI) and *p*-values were also presented. The final models were assessed using the concordance (c) statistic for discriminative ability (or area under the receiver operating characteristic [ROC] curve), the Hosmer-Lemeshow goodness-of-fit test to evaluate the calibration (a *p*-value over 0.05 would show an acceptable adaptation), and the Nagelkerke R^2^ to present the proportion of explained variance [40]. A probability level of 0.05 was employed for all statistical tests. IBM SPSS version 27.0 (IBM, Armonk, NY, USA) was used.

Qualitative analyses were conducted using content analysis with similar responses clustered together reflecting the frequency and, then, the interpretation of responses collated [41]. To minimize any bias in the analysis, given that the first author was also the lead author of STARS-p (and the STARS-p training), two co-authors (KK, and KKr) were involved in categorizing responses into meaningful themes and categories of responses with similar meanings. Differences in coding were discussed and finalized with mutual agreement. Direct quotes were used to illustrate categories of responses. Considering the similar content of the two open-ended questions, responses to these two questions were collapsed for analysis and coded accordingly (i.e., coding and analyses were not specific to each question).

## 4. Results

### 4.1. Participants

In total 70 participants completed the personal background parts of the survey. Of these, 67 participants completed the entire survey and were included in the final analyses. A summary of participant demographic and clinical characteristics is presented in Table 1. Participants were mostly female (80.6%) and were MHPs with a formal qualification (MHP-wQ) (68.7%). Their average age was 43.2 years (SD = 10.3) and they had on average 9.5 (SD = 7.8) years of experience in suicide prevention. The majority saw a suicidal client at least monthly (82.1%) and had experienced a suicide attempt of a client (83.3%), with over a quarter (27.4%) having lost a client by suicide. Furthermore, the majority had received informal training (supervision) at least monthly (92.5%) and 35.8% had received some formal training (workshops) since they completed the STARS training.

### 4.2. Use of STARS-p and Reasons for and Perceptions of Use

Most of the participants (80.6%) had used the entire STARS-p at some point in their work with clients since their original STARS training. However, less than half (44.7%) frequently (75–100% of time) administered the entire STARS-p in their work with clients in one administration (Table 2). That is, administration of Parts A, B, and C, as well as safety planning and clinical notes within the administration interview. Part A (suicide enquiry items) was the most frequently administered section of the STARS-p (92.2%), with the majority (68.8%) using it 100% of the time. Part C (protective factors) was the second most frequently used section; 56.1% of participants used it 100% of the time and a further 24.2% used it 75% of the time. Part B (psycho-social risk factors) was also frequently used by 71.2% of participants, while nearly three quarters (74.2%) frequently administered the Safety Planning section. Notably, the Clinical Notes section was the least frequently used section; 57.6% used it at least 75% of time.

The most common reason for using the STARS-p was ‘direct verbalized suicide ideation’ (Table 3), followed by ‘observed mental illness symptoms/signs’, ‘comments of the person’ and ‘specific mention in the referral’ (all endorsed by 76.1%). The least frequent reason was ‘I always conduct a risk assessment—mandatory in my organization’ (31.3%).

Over half (57.9%) of participants perceived the STARS-p as easy (moderately or extremely easy) to administer, while over three quarters (78.1%) perceived that their clients felt validated (understood) when using STARS-p. The large majority of participants (90.6%) were confident in the usefulness of information collected with the STARS-p, while 79.7% were confident that its use informed needs-based priority areas to mitigate clients’ suicidality, and 84.4% considered it to be a client-centered approach to determining a client’s suicidality.

### 4.3. Associations between Use of STARS-p, Professional Characteristics, and Perceptions of STARS-p

Correlation analyses showing associations between use of STARS-p and perceptions of the protocol and professional characteristics of the sample can be cited in Appendix A.

Forward stepwise likelihood ratio logistic regression models were used to identify variables independently associated with use of entire STARS-p and its parts (frequent use = 100% or 75% of the time; infrequent use = never to 50% of the time). As shown in Table 4, frequent use of the entire STARS-p was significantly associated with greater ease of administration and male gender (R^2^ = 0.51). Similarly, use of Part A was associated with greater ease of administration and prior client suicide attempts (R^2^ = 0.35). Frequent use of Parts B and C were associated with greater confidence that the STARS-p informed needs-based priority areas to mitigate clients’ suicidality (R^2^ = 0.20; 0.30 respectively). Similarly, frequent use of Safety Planning and Clinical Notes sections were associated with greater confidence that the STARS-p informed needs-based priority areas. In addition, frequent use of Safety Planning was associated with perceptions that STARS is client-centered, whereas frequent use of the Clinical Notes section was associated with male gender and greater ease of administration (Table 4).

### 4.4. STARS-Trained Practitioners’ Perceptions of STARS-p and Improvements for Practice

Fifty (75%) participants provided additional views on the use of STARS and/or improvements for practice. The content analysis of responses to the two open-ended questions yielded multiple categories which reflected three themes; namely, facilitators or advantages to using STARS-p in practice, barriers or factors discouraging use of STARS-p, and suggestions to improve use of STARS-p in practice. Table 5 presents the themes, categories, and illustrative quotes. Regarding *facilitators or advantages to using STARS-p in practice*, the five categories (summarizing 58 responses) characterized the STARS-p and/or training as practically helpful, flexible for tailoring, client-focused, comprehensive, and a “mindset” rather than just an assessment. The four categories reflecting *barriers or factors discouraging use of STARS-p* (44 responses) highlighted time constraints, client-related factors, cumbersomeness, and organizational elements. The third theme, *suggestions to improve use of STARS-p in practice* (20 responses), was characterized by three categories including: changes to STARS-p (sections), need for ongoing training and need to practice using STARS-p.

## 5. Discussion

This study aimed to examine the use and application of STARS-p in STARS-trained practitioners over the 12–24-month period post STARS-training. Our first aim was to examine STARS-trained practitioners’ use, perceptions, and reasons for use of the STARS-p since training. Based on earlier findings concerning STARS-p use [10,34] we hypothesized that the majority of participants would not use the entire STARS-p, due to its length, and that they would most commonly utilize the protocol when clients expressed suicidality. We identified that at 12–24 months post original STARS training, most participants (80.6%) had used the entire STARS-p at least once in their practice. Further, as expected, less than half (44.7%) frequently (75–100% of time) used the *entire* STARS-p. Rates of frequent use were higher for Parts A, B and C, with 84%, 71% and 82% participants using these 75–100% of the time, respectively (further details are outlined below). Over half of the participants (58%) perceived the STARS-p as easy to administer, three quarters reported that their client felt validated by the administration process, and a majority of participants reported feeling confident in using STARS-p for identifying priority needs for informing care responses (80%) and perceived that the protocol was client-centered (84%).

### 5.1. STARS-Trained Practitioners’ Use of STARS-p Sections

Our findings revealed that Part A (suicide enquiry) of the STARS-p was used most frequently in practice; with nearly 70% of the sample reporting that they used it *all of the time*. In STARS training, completion of the suicide enquiry is emphasized as being the most critical component of undertaking SRA, especially for presentations of heightened client suicidal distress. The published guidelines within the inside cover of the STARS-p also emphasize the critical need for completion of Part A for such presentations. The related competency taught in STARS-p training focuses on exploration of clients’ past and present thoughts, intent, plans and behavior, Although this competency is acknowledged to be most important in SRA, yet it is the least well done by clinicians [42]. This finding may highlight the potential positive impact of the STARS-p training on participants’ use of Part A in practice in the longer-term (12–24 months after original STARS-p training).

An unexpected but positive finding was that the ‘Client ratings of concern’ function of the STARS-p, was used frequently (75–100% of the time) by 74% of participants in Part A, and by 70% of participants in Part B. Client ratings of concern, or subjectively perceived priority factors contributing to suicidality, are a critical indicator for informing safety planning and management responses, which is strongly emphasized in the STARS training. The ‘Therapist confidence rating’ function (therapist confidence in the clients’ own ratings about what is a contributing concern or factor; in Parts A and B) was used frequently (75–100% of the time) by a smaller proportion (62%) of participants during administration of STARS-p. This rating is designed to encourage transparent discussion and open dialog between client and clinician, particularly when diversion between client and clinician ratings exists. It may be that therapist confidence ratings are not deemed as a priority for SRA considering ‘time pressures’, as frequently cited as a barrier to use of the entire STARS-p [34].

Given psycho-social needs-based assessment is a central feature of STARS-p, it was affirming to find that Part B was frequently used by 71% of the sample (75–100% of the time). Furthermore, half of the sample used safety planning *all of the time* in their administration of STARS-p, and almost three quarters of the sample used safety planning *at least 75%* of the time. This finding highlighted the importance that participants placed on safety planning and its integral role in SRA. Yet, in another study on MHP’s capabilities, most of the sample were unfamiliar with safety planning or did not use it [43], so the current findings were promising.

We found Part C (protective factors) to be frequently administered by 82% of the sample (75–100% of the time). The importance of protective factors as a component in the safety planning process is emphasized in STARS-p training, which may have influenced the high frequency of use of this section. Client protective factors are commonly referred to in strength-based approaches for working with those in suicidal distress [44,45].

An unexpected finding on use of STARS-p was MHPs’ lack of use of the Clinical Notes section. Only 29% of the sample frequently used the documentation section (75–100% of the time). This may reflect either a lack of demonstrated competency in Documentation and Duty of Care (module 4 of STARS training) [10] or that MHPs employed alternative methods to record their suicide specific documentation. In any case, this finding highlighted a need to further emphasize the importance of documentation in duty of care responsibilities when working with those in suicidal distress in future STARS training.

### 5.2. Main Reasons for Using STARS-p in Practice, and Perceptions of STARS-p

As hypothesized, direct verbalizations by the client about suicide was the main reason for MHPs administering STARS-p in practice. However, more than 60% of all the listed reasons were endorsed by more than 70% of participants. While each of the listed items are credible reasons for initiating an SRA, the STARS training specifically points to direct client verbalizations and symptoms of distress (or mental illness, such as depression, agitation etc.) as important indicators for administering STARS-p. These findings supported the benefits of STARS training for practitioners to gauge the need to administer STARS-p, appropriately, when faced with clients in suicidal distress.

Not surprisingly, given the willingness of participants to undertake this study, we found that most had positive perceptions about the STARS-p. Between three quarters and 90% of practitioners had positive perceptions about STARS-p being client-centered, supporting the client to feel validated, and providing confidence in the information gathered for informing care. However, approximately 40% perceived it as being difficult to administer in their practice. Findings from our qualitative analyses (see Section 5.4) provided potential explanations for this finding.

### 5.3. Use of STARS-p and Associations with Professional Characteristics and Perceptions of the STARS-p 12–24 Months after Training

Our hypothesis that greater formal and informal training would be independently associated with administration of the entire STARS-p (or sections of the STARS-p) was not supported. Given the existing evidence, discussed in other prior research [10], about the positive impacts of training on practitioners’ capabilities in SRA (i.e., impacts on attitudes, perceived capability, and knowledge in SRA), it was expected that previous training and supervision would influence the administration and use of STARS-p in practice. In this current study, however, only 36% of our sample had engaged in formal training since their original STARS training (12–24 months ago). This meant that the opportunity to refresh knowledge and skills associated with STARS-p administration was either limited or absent for the majority of STARS-p licensed users in our study. Further, while nearly 70% had engaged in informal training (e.g., monthly supervision) since their original STARS training, this informal education may not have been focused on STARS-p administration processes, designed to enhance frequency of use.

Interestingly, we found that administration of the entire STARS-p was independently associated with both male gender and perceiving STARS as easy to administer. Although, it was unclear why males (versus females) in our study were more likely to administer the entire STARS-p, previous research by Crowley [46] found that male practitioners (and those with more intensive CAMS training) adhere more closely to the underlying philosophy of CAMS. Further, given that there were no major gender related differences in our results, this might be an indication of the benefits of a standardized, structured professional judgement approach in administering STARS-p. Similarly, practitioners who perceived STARS-p as easy to administer were more likely to frequently administer Part A. Further, those who more frequently administered Part A, were also more likely to have had a client attempt or die by suicide. Those who have already experienced client suicide loss and/or attempt are known to be more cautious in their future SRA practice [47] and operate with greater adherence to treatment protocols [30,48].

Practitioners’ greater confidence in use of STARS-p for informing needs-based priority areas to mitigate clients’ suicidality was independently associated with frequent use of Parts B and C, the Safety planning and Clinical notes sections. These findings suggested that the content covered in Parts B and C, and the functions of safety planning and documentation of commensurate care (informed specifically by Parts B and C), were well aligned with the main aim of STARS; to inform needs-based priority foci for subsequent care [29]. Interestingly, the finding that practitioners’ perception of STARS as client-centered was associated with frequent use of Safety planning, might be reflective of the emphasis on individualization and tailoring of safety planning [49,50,51]. This principle is heavily emphasized in the STARS training, concerning collaborative development of individualized safety plans. Finally, male gender and perceptions of the protocol being easy to administer were independently associated with frequent use of the Clinical notes section of STARS-p. It was not surprising that this written section of the STARS-p, which can be time-consuming, was completed mostly by those who found it easy to use. Documentation processes can be complex and time-consuming, particularly when suicidality is foreseeable; so, perceiving this administration process as easy was logically more likely to lead to its more frequent practice.

### 5.4. Perceptions of STARS-p and Improvements for Practice

Qualitative analysis of STARS-trained MHPs’ perceptions of the STARS-p and training and suggestions for improvements, resulted in the following three main themes: facilitators or advantages to protocol use, barriers or factors discouraging protocol use, and suggestions to improve protocol use. Importantly, the key categories under these themes depicted ‘real-world’ understandings about STARS-p application and future developments.

Nearly half of the responses characterized facilitators and advantages of using STARS-p, endorsing the practical elements of the protocol, and the ability to use it flexibly, allowing practitioners to tailor the administration to different client presentations. Specifically, reference to the protocol as practically helpful for assessments (such as the psycho-social needs-based component with more complex clients, and in post-discharge/aftercare settings), and the ability to use different parts of the protocol were considered valuable. The client-centered focus of the assessment administration process of STARS-p was also emphasized, which reinforced the findings of our earlier pilot investigation of STARS-p (2015 Edition: [34]). Further, the view of the protocol as a ‘mindset’ reaffirmed the philosophical underpinnings of the STARS training and approach.

The second theme regarding barriers or factors discouraging use of the protocol, predominantly depicted time constraints; namely, the time-consuming nature of STARS administration. Practitioners noted that while the protocol was comprehensive, client sessions in the community mental health (private practice or otherwise) setting were limited in duration (50–55 min), which hindered completion of the ‘entire’ protocol in one session. Client-related factors were also seen as a challenge to administration of STARS-p, particularly in relation to cultural differences (e.g., acknowledgement of preference for ‘yarning’ as opposed to ‘pen/paper response’ format for Aboriginal and Torres Strait Islander populations).

Finally, and key to informing the implications for future STARS-p refinement, were MHPs’ suggestions to improve protocol use. For example, expansion of space for documentation of client responses (e.g., Safety planning section) was indicated, as were suggestions for abbreviated versions of the protocol (with priority assessment areas to be administered at least in the first instance/session). Preference for a digital version of STARS-p that would allow repeated use and updating of client information at subsequent assessments, concurred with clinician feedback provided on the 2015 Edition of STARS-p [34]. Other assessment and management protocols recently commenced testing of the digital counterpart version of original hard copies with some promising results [52]. The need for ongoing training and practice using STARS-p was suggested to enhance frequency of use over time and confidence in use respectively. This user-insight provided invaluable feedback for the authors around future planning for ongoing refresher STARS training and support for organizations who invest in the initial training.

### 5.5. Limitations

The study had several limitations, including the inability to prospectively follow-up participants and link data from the original STARS training. The present participant sample reflected only a modest portion of the broader group of MPHs trained in the STARS-p and it was not possible to verify their use of the STARS-p over the 12–24-month period. More generally, it is not known how representative this sample is of the originally trained cohort [10]. Nonetheless, our sample size was adequate to detect mainly moderate sized correlations between demographic and clinical characteristics and perceptions and use of the STARS protocol.

### 5.6. Implications

Our results provide important future directions concerning STARS-p modifications, as well as STARS training. First, we intend to investigate the potential for a briefer version or ‘two-parts’ of the STARS-p interview, in acknowledgement of the tendency for more frequent use of Part A and Safety planning sections, particularly for acute presentations of suicidal distress. Further, developing a digital based version of the protocol for more efficient data entry and synthesis of information may help to reduce ‘cumbersome’ note taking, as well as integrate with existing organizational digital client records. Increased refresher training opportunities should also be offered to enhance capability, confidence, and ease of administration of the protocol. Finally, future research should explore the perceptions of those with a lived experience of suicide regarding acceptability of language and terminology of the STARS-p, as well as clients’ perceptions and experiences of its administration.

## 6. Conclusions

The current study investigated how STARS-trained MHPs utilize STARS-p in the real-world with their clients over the longer-term (i.e., 12–24 months after training). The key findings were that most participants did not frequently use the entire protocol but rather flexibly used different sections. Frequent use of either the entire protocol or different sections was independently related to male gender, perceived ease of administration and confidence in the use of the protocol. Taken together with the qualitative results, these findings provide important directions for developments of STARS-p and training in practice.

## Figures and Tables

**Figure 1 ijerph-19-11324-f001:**
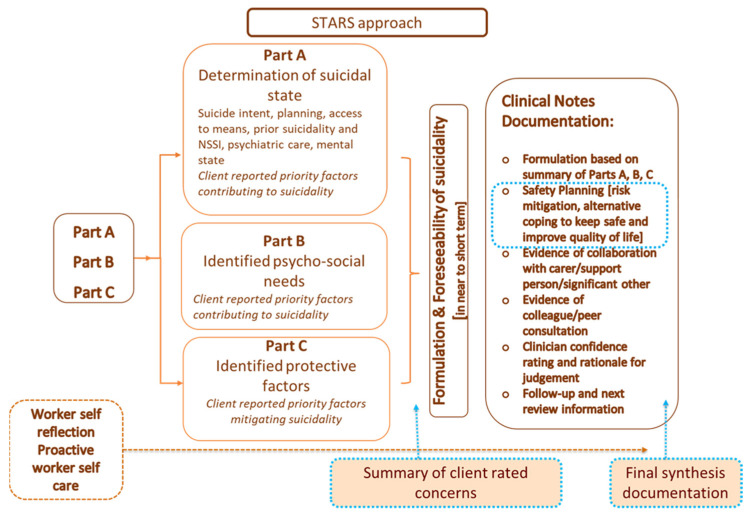
The STARS protocol (from Hawgood et al. [10]).

**Table 1 ijerph-19-11324-t001:** Description of participants (N = 67).

**Gender**	N	%
Male	13	19.4
Female	54	80.6
**Work status**	N	%
Full time	45	67.2
Part time	20	29.9
Casual	2	3.0
**Role**	N	%
MHP-wQ	46	68.7
* MHP-woQs	21	31.3
**Education**	N	%
High school or TAFE	5	7.5
Undergraduate	14	20.9
Postgraduate	44	65.7
Professional doctorate/PhD degree	4	6.0
**Received any FORMAL training in suicide prevention/management SINCE STARS training**	N	%
No	43	64.2
Yes	24	35.8
**Received INFORMAL training**	N	%
Weekly	2	3.0
Fortnightly	14	20.9
Monthly	46	68.7
Several times per year or less	5	7.5
**Contact with suicidal persons**	N	%
Daily	5	7.5
Several times per week	10	14.9
Weekly	9	13.4
Several times per month	20	29.9
Once per month	11	16.4
Multiple times per year	12	17.9
**Loss of client by suicide (missing = 5)**	N	%
No	45	72.6
Yes	17	27.4
**Suicide attempt of client (missing = 1)**	N	%
No	11	16.7
Yes	55	83.3

* Note: MHP-wQ = mental health practitioner with formal MH qualification; MHP-w0Q = mental health practitioner without formal MH qualification (e.g., trained peer or support workers).

**Table 2 ijerph-19-11324-t002:** Use of entire STARS-p and sections of the protocol.

Use of STARS-p	100% of Time		75% of Time		50% of Time		25% of Time		Never		Missing
	N	%	N	%	N	%	N	%	N	%	
Administering the entire STARS-p (Parts A, B, C, safety planning and documentation sections)	9	13.4	21	31.3	9	13.4	15	22.4	13	19.4	0
Administering PART A (Suicide Enquiry items)	44	68.8	10	15.6	1	1.6	4	6.3	5	7.8	3
Client rated concerns	27	41.5	21	32.3	3	4.6	6	9.2	8	12.3	2
Therapist rated concerns	18	27.7	22	33.8	9	13.8	7	10.8	9	13.8	2
Administering PART B (Psycho-social risk factors)	23	34.8	24	36.4	9	13.6	4	6.1	6	9.1	1
Client rated concerns	19	28.8	27	40.9	7	10.6	5	7.6	8	12.1	1
Therapist rated concerns	14	21.2	27	40.9	9	13.6	8	12.1	8	12.1	1
Administering PART C (Protective factors)	37	56.1	17	25.8	6	9.1	1	1.5	5	7.6	1
Administering the SAFETY PLANNING section (within the Clinical Notes section)	33	50.0	16	24.2	6	9.1	4	6.1	7	10.6	1
Administering the DOCUMENTATION NOTES section (within the Clinical Notes section)	19	28.8	19	28.8	9	13.6	6	9.1	13	19.7	1

**Table 3 ijerph-19-11324-t003:** Reasons for the use of STARS-p.

Reason for You to Use the STARS-p with a Client *:	Yes Responses	
	N	%
Direct verbalized suicide ideation	63	94.0
Observed mental illness symptoms/signs (e.g., apathy, withdrawal etc)	51	76.1
Specific mention in the referral	51	76.1
Comments of the person	51	76.1
Circumstances of the person (presenting problem)	49	73.1
Client reported behavioral changes	48	71.6
Emotional/distress observed in voice (e.g., anger, anxiety etc)	45	67.2
Reported life stressors/crises (e.g., financial, custodial, relationship etc)	44	65.7
Observation of the person	43	64.2
Changes in (increase) or habitual drug and alcohol use	42	62.7
Indirect (passive verbalization) ideation	41	61.2
I always conduct a risk assessment—mandatory in my organization	21	31.3

* Multiple responses from participants were possible.

**Table 4 ijerph-19-11324-t004:** Multiple logistic regression final models (likelihood ratio) of association with use of the STARS-p.

	OR	95% CI		*p*-Value
		Lower	Upper	
**Full STARS (n = 64)** (Nagelkerke R^2^ = 0.513)				
Gender (male vs. female)	17.26	2.16	138.12	0.007
Ease of administration of the STARS-p	20.43	4.11	101.51	<0.001
**Part A (n = 61)** (Nagelkerke R^2^ = 0.349)				
Ease of administration of the STARS-p	8.94	1.76	45.40	0.008
Suicide attempt of client (yes vs no)	7.03	1.29	38.31	0.024
**Part B (n = 63)** (Nagelkerke R^2^ = 0.203)				
Confidence in the use of STARS-p	8.20	2.05	32.76	0.003
**Part C (n = 64)** (Nagelkerke R^2^ = 0.301)				
Confidence in the use of STARS-p	13.71	3.08	61.05	0.001
**Safety planning (n = 64)** (Nagelkerke R^2^ = 0.320)				
Confidence in the use of STARS-p	5.09	1.15	22.46	0.032
STARS as a client-centred approach to determination of client’s current suicidality	6.77	1.29	35.51	0.024
**Notes (n = 63)** (Nagelkerke R^2^ = 0.440)				
Ease of administration of the STARS-p	4.91	1.34	17.97	0.016
Confidence in the use of STARS-p	6.39	1.02	40.00	0.048
Gender (male vs. female)	12.90	1.19	140.37	0.036

**Table 5 ijerph-19-11324-t005:** Qualitative content-analysis of open-ended questions: Perceptions of STARS-p in practice and, comments and suggestions for STARS-p and training.

Questions (1 and 2 Collapsed)	Facilitators or Advantages to Protocol Use (58 Responses)	Barriers or Factors Discouraging Protocol Use (44 Responses)	Suggestions to Improve Protocol Use (20 Responses)
**1. Perceptions of long-term protocol use** (n = 50) **2. Comments & suggestions for protocol & training** (n = 36)	**Practically helpful for assessment** (n = 39) *“I really value the program as it enhances and adds value to what I do in assessing clients (thanks)”* *“STARS is perfect for our complex clients. We are encouraged to do psycho-social needs-based responses, so....”* *“The protocol is great for my colleagues and I because we get referrals after discharge from hospital and staff at the ED rarely do a psycho-social assessment of needs and rarely ask the client what they think is their most important issue”* *“Despite hearing others’ refer to the length and intensity of using STARS, I think the tool does reduce clinicians jumping to action and encourages us to really sit with someone experiencing thoughts of suicide to understand their experience and work together on a plan”* *“STARS training was fabulous”* *“I found the training was very useful”* **Flexible for tailoring** (n = 7) *“I often have the whole session to administer STARS if it is an ‘intake’ session, but if not, I get through Part A and safety planning which is what we were told in the training—plus I would do documentation”* *“Part B is perfect for just informing the support plan. The documentation section helps with duty of care requirements in our organization”* **Client-focused in administration** (n = 8) *“Parts A, B and C and the safety planning section of STARS is really useful and helps to guide the risk assessment in a way that is still client focused”* *“Very client-centered, have found it very helpful”* *“Definitely client focused—helps me to confirm my observations and client intent”* **Comprehensive** (n = 2) ***“****I know that the STARS is a comprehensive interview—it takes time, but it is necessary and with COVID we need more reliable data like this”* **“A mindset” rather than just an assessment** (n = 2) *“For me it is the approach and mindset rather than the form that is the vital skill”*	**Time constraints** (n = 27) *“As a psychologist, usually limited to 50–55-min sessions, it is difficult to complete the whole STARS-p and still provide the client with therapy/treatment and meet the clients’ expectations to sufficiently engage them in therapy (lest they disengage at a point they are suicidal)”* *“Time constraints in the service make these questions difficult to answer fairly. Is there any way it could be broken down, so it was able to be used more fully in a service with limited client contact time?”* **Client-related factors** (n = 12) *“Aboriginal and Torres Strait Islander community prefer to yarn and not use pen and paper”* *“[However] …Some of my more acute clients would require two sessions to administer the whole protocol, unless I have time for a break and can therefore administer within the one session”* *“It is sometimes hard to use STARS with all clients. My biggest challenge is using it with people who are completely non-talkative. But when I ask them what they think the factors are they do open up”* *“I find the need for a risk assessment to be a difficult document as it takes away from me giving the client my whole attention—instead there is a heap of papers between us and it does not feel genuine to me”* **Cumbersomeness** (n = 2) *“I find it to be too lengthy”.* *“I personally find it difficult to use something so structured—something that would really need to be printed and gone through one step at a time”.* **Organizational (not mandated/required to use)** (n = 3) *“I do not use the STARS-p; I came to hear about it at the time because I was involved in facilitating a course on suicide prevention at xxx University. We wanted to learn about STARS for this reason, and I was mandated to go and participate in the training”* *“We are no longer required to use STARS; I use other tools”.* *“Unfortunately, I did not use it afterwards (after training).”*	**Changes to STARS-p (sections)** (n = 9) *“It would be great if the Safety Planning section was expanded with a lot more space. Similarly, this would be helpful in the sections that ask about method, and previous attempts, so there is more space to document the intricacies of what is reported”.* *“A more time conscious, abbreviated version of the tool would be wonderful given that a 51-min session only enables use of some elements of the protocol, while also conducting the necessary safety planning protocols that my workplace and practice requires”* *“I think it if had a digital version that succinctly put information together to save time, I would use it more”.* *“A digital form would be helpful—so that we can update and modify by saving a new copy each new file entry for the same client”* **Need for ongoing training** (n = 6) *“I think that people in my service need to get ongoing training in STARS. They have done the first training but then ‘give up’ on using the protocol and simply jot down assessment data in their charts. It is always so busy and staff do not seem to get that if you invest at the beginning, you will be even less likely to stress about needing more time down the track”* *“We need more refresher training—I did this, and it was excellent”* **The need to practice using STARS-p** (n = 5) *“The more I use it the more confident I get knowing I have gathered all the information”* *“Good instrument—we need to practice/develop better understanding in supervision sessions”*

## Data Availability

Data can be requested from the corresponding author.

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
