# Peer review of "Long-Term Use and Application of Systematic Tailored Assessment for Responding to Suicidality (STARS) Protocol Following Original Training"

_ijerph, 2022, doi:10.3390/ijerph191811324_

Round 1

Reviewer 1 Report

This paper aimed to assess the long-term use and application of the Systematic Tailored Assessment for Responding to Suicidality protocol (STARSp). The main goal of the study was to examine how mental health practitioners (MHPs) utilized STARSp 12-24 months post-training to investigate how useful this tool is for preventing suicide and to evaluate if there are improvements to the protocol that could be made.

Specifically, the authors focused on the MHP’s perception/reasons for using the STARSp in their clinical practice as well as the demographics of MHP that used the STARSp. In addition, the authors assessed the associations between MHP’s demographic and clinical characteristics to better understand the utilization of STARSp. Using this information, the authors were able to collect data that allowed them to conclude that STARSp as a whole or its parts is frequently used. Advantages and barriers to using STARSp in practice will aid in informing future STARSp developments and training.

The use of this tool and its clinical relevance is delineated in this paper. It is helpful to know the history and reasoning behind utilizing STARp for suicidal patient populations. Also, to know what is available in the market. In addition, the authors roughly describe the STARS approach which is helpful as a background for the reader as the reviewer (reader) who has no access to the actual program itself. However, I would recommend condensing this part and restricting it to the discussion of STARSp. 

The findings of the study support the hypothesis the authors created in the introduction. For example, the authors hypothesized that direct verbalization of suicidality by the client would be the main reason for MHP’s utilizing STARp. This was corroborated by the results from the survey that was administered for the study.

However, it is stated and supported that certain hypotheses were not confirmed. For example, the authors hypothesized that greater formal and informal training would be independently associated with the administration of the entire STARSp, however, their results did not support this hypothesis.

This protocol uses subjective assessment which necessitates training of the staff who uses them.  However, the data suggest that despite the training only 45% of practitioners used this tool in its entirety. This is explained in the discussion under limitations, where time constraints are the main reasons for not using STARSp.

The methods along with the results sections were clear, however, the section explaining the rejected hypothesis was confusing to follow. In addition, Tables 2, 4, and 5 contained data that could be placed in a supplemental section. The information provided in these tables does not provide clinical relevance or strengthen the author's argument.

Furthermore, table 6 repeats information that is already included in the text. It is suggested that this table either be deleted or the text be deleted so that there is no redundancy.

The authors did not elaborate on what their thoughts were regarding the rejection but instead used supporting information from another study to rationalize this thought.

Overall, the conclusions made by the paper are supported by the data presented in the article, however, most sections need to be reworked so that the paper is succinct.

Author Response

Please see attachment: Response to Reviewer 1. Also please see Updated manuscript with track changes indicating amendments as requested, as well as ESM1 file. 

Reviewer 2 Report

An important finding was the following:

Results: 67 participants (81% female, Mage = 43.2, 24 SD=10.3) were included in analyses. 80.6% of participants had used the entire STARS-p at some 25 time-point in their practice; less than half (44.7%) frequently used the entire STARS-p. Parts A, B 26 and C were used frequently in suicide risk assessment (SRA) by 84%, 71% and 82% of participants,  respectively.

What was confusing to me was how this resulted in only 44% using all three parts--clarification in the paper would be helpful.

Author Response

Please see the attachment - Response to Reviewer 2, as well as updated manuscript with track changes and ESM1 now included. 

Round 2

Reviewer 1 Report

Thank you for incorporating the suggested edits. This current version of paper is easy to read and seems focused. 

Author Response

Dear Reviewer, Thank you for your supportive comments, and we are glad to know that there are no more edits required for publication. 

Kind regards

jacinta